

# In-depth analysis of a discrete $p$ model

Uwe Saint-Mont, Nordhausen University of Applied Sciences

August 29, 2019



# Abstract

Towards the end of the last century, B. Mandelbrot saw the importance, revealed the beauty, and robustly promoted (multi)fractals. Multiplicative cascades are closely related and provide simple models for the study of turbulence and chaos.

For pedagogical reasons, but also due to technical difficulties, continuous stochastic models have been favoured over discrete cascades. Particularly important are the $\alpha$, the $\beta$ and the $p$ model (Lovejoy and Schertzer (2013), Chapter 3, de Wijs (1951, 1953)). It is the aim of this contribution to introduce original concepts that shed new light on the latter paradigmatic cascade and allow key features to be derived in a rather elementary fashion.

To this end, we introduce and study a discrete version of the $p$ model which is based on a new kind of sampling. Technical machinery can be kept simple, therefore formulas are explicit, proofs extend standard arguments, and potential extensions are numerous. Thus the proposed line of investigation may enrich and simplify received multifractal analyses.

**Keywords**: p model, binomial cascade, multifractals, sampling, law of large numbers



# 1  Introduction

Cascades are straightforward and excellent models for divergent phenomena. That is, given some point, the mass concentrated at this point is distributed to a number of descendants. Straightforwardly, with a single starting point (the root) and repeated local bifurcations (all governed by the same mechanism), one obtains a tree-like and self-similar structure that can often be extended to a reasonable (multi)fractal limit (Shynkarenko 2019).

Early examples of fractals were provided by, among others, mathematicians Weierstraß, Cantor and Peano. Later, upon studying dynamic systems, chaos and turbulence, physicists found similar patterns. Mandelbrot (1982, 1997, 1999) obtained a first synthesis when he established a strong link between fractal geometry and its applications in the sciences (physics and economics in particular) which has since been extended to "multifractal methodology" (Salat et al. 2017), general "critical phenomena" (Sornette 2007), and asymptotic theory (Kendal and Jørgensen 2011).

Moving from the objects involved to the processes generating them, cascades have come into focus (Schertzer and Lovejoy 2011) only recently. On the one hand, they are quite common. On the other hand, they are also a "key idea" conceptually (Lovejoy (2019), p. 76). That is, although a cascade's components are rather primitive, they can easily be adapted to observable phenomena:

The basic building block - a type of fork-like structure - can be chosen appropriately, the propagation mechanism may be deterministic or stochastic (the original $p$ model vs. most other models), scales (and scale invariance) are closely related, and the approach works in spaces of (almost) any dimension. Endowed with an abundance of nonlinear processes, geophysics has been especially productive in this respect (see, in particular, Mandelbrot (1989), and Lovejoy and Schertzer (2007)), with contributions ranging from the atmosphere (climate and weather), wave dispersion and topography to geology and mining (e.g., Serinaldi (2010), Lovejoy and Schertzer (2013), Agterberg (2019)).





Pioneering work dates back to the first half of the 20th century, in particular to Richardson (1922) and Kolmogorov (1941). A little later, de Wijs (1951, 1953) established the basic $p$ model (see also Mandelbrot (1974), p. 329): For $n = 0$, start with the uniform distribution on the unit interval. Next, the proportion $1 - p$ is uniformly distributed on the interval $(0, 1/2)$, and the proportion $p$ is uniformly distributed on the interval $(1/2, 1)$. In the same vein, one splits the masses further (locally), i.e., mass $(1 - p)^2$ to the interval $(0, 1/4)$, mass $(1 - p)p$ to the interval $(1/4, 1/2)$, mass $p(1 - p)$ to the interval $(1/2, 3/4)$, and mass $p^2$ to the interval $(3/4, 1)$, etc. It is well-known that the corresponding limit distribution function, with the exception of $p = 1/2$, has no density (Salem 1943).

Curiously enough, Mandelbrot (1999), p. 87, says that the $p$ model appeared "in an esoteric corner of mining engineering science." However, if one thinks about it, ores are the result of an enrichment process. Such a process may be modelled as a sequence of binary decisions, i.e., a cascade that is biased in favour of some mineral. Owing to Salem's result, it is to be expected that the limit of such a process should be some kind of fractal, involving a certain amount of polarization (a natural mineral deposit vs. dead rock, say). For a graphic example see Hill (1999).

Since the basic building block used in the $p$ model is a binary bifurcation (each point bequeaths its mass to two descendants with proportions $p$ and $1 - p$, respectively), the corresponding cascade should be named after *Bernoulli*. Unfortunately, the terms 'binomial cascade' (and 'binomial measure') have caught on in the literature, since such a process traditionally yields a binomial distribution. This article shows that this need not be the case.

It is instructive to compare a Bernoulli cascade with the classical Galton board: Each ball running down the board also makes a binary decision at every step. However, since each bifurcation is counterbalanced immediately afterwards - with exactly two possible paths the ball may take *merging* at another point - extreme imbalances are rather unlikely, and a smooth density occurs at the bottom of the board. More precisely, one starts with unit mass at a single point, and a Bernoulli random variable $B(p)$ governing





⁷⁶ the binary decision of moving left (failure) or right (success). $k$ successes in $n$ trials can

⁷⁷ only occur if $k - 1$ successes in $n - 1$ trials are followed by a success, or if $k$ successes

⁷⁸ in $n - 1$ trials are followed by a failure in the last trial. Thus paths split and merge

⁷⁹ successively, leading to Pascal's triangle and the corresponding binomial distribution

⁸⁰ $B(n, p)$. Asymptotically, one gets a smooth normal distribution with most of the mass

⁸¹ close to some centre of gravity, which was first proved by De Moivre and Laplace, and

⁸² later extended to the *central* limit theorem (CLT, note the name).

⁸³ Quite obviously, there are two kinds of opposing 'forces' at work: On the one hand,

⁸⁴ bifurcations split some material and cause variance. Thus they dissipate matter effi-

⁸⁵ ciently, but may also concentrate it in some places (veins of gold, for instance). On

⁸⁶ the other hand, 'mergers' amass materials and eliminate variance. Combining material

⁸⁷ from various sources, they also blend their input (errors, for instance) efficiently. A

⁸⁸ particular important kind of merging is *averaging* which, at least typically, leads to

⁸⁹ continuous unimodal densities.

⁹⁰ In order to achieve some kind of polarization, it seems a good idea to avoid 'mergers' and

⁹¹ to look for distribution functions that are not differentiable, i.e., cascades in general,

⁹² and the classical $p$ model in particular. Beyond that, an elementary *discrete* cascade

⁹³ with similar properties also in the finite case would be even more appropriate. After

⁹⁴ giving a more abstract motivation in the next section, such a process will be defined

⁹⁵ and studied rigorously throughout the rest of this article.

⁹⁶ The basic idea is to establish a rather strong 'force' that is able to separate different

⁹⁷ classes of object. More precisely, starting with two distinct populations *exponential*

⁹⁸ *sampling* is able to prevent them from merging. It turns out that the corresponding

⁹⁹ deterministic cascade forms 'threads' that interweave in a systematic way defining a

¹⁰⁰ 'multiplicative triangle.' The corresponding distributions are discrete versions of well-

¹⁰¹ known continuous distributions with 'sewn-in' binomial components (section 3).

¹⁰² Expected values and variances are derived in section 4, asymptotic properties are dis-

¹⁰³ cussed in section 5, and section 6 is devoted to populations with finite variances. In




¹⁰⁴ particular, the variance can be decomposed into several components. Finally in section

¹⁰⁵ 7, we give a number of potential extensions.

¹⁰⁶ It might be mentioned that the new mathematical structures are as basic as the bi-

¹⁰⁷ nomial distribution $B(n, p)$ and its siblings. Indeed the finite *Weaver distributions*

¹⁰⁸ $W(n, p)$ and their limit $W(p)$ are straightforward consequences of a Bernoulli cascade.

¹⁰⁹ Technically, the crucial difference is a slightly more sophisticated way of sampling

¹¹⁰ that 'augments' Pascal's triangle to a multiplicative pattern. The latter 'triangle' is

¹¹¹ equivalent to local Bernoulli bifurcations and brings out the fractal nature of zero-one

¹¹² decisions.

## ¹¹³ 2  Theoretical motivation

¹¹⁴ Traditional statistics rests on several main theorems, in particular CLTs and laws

¹¹⁵ of large number (LLN). Given an iid sequence $X_1, X_2, \dots$ of random variables, the

¹¹⁶ basis of Frequentist statistics is some LLN, i.e., the convergence of $\bar{X}_n = S_n/n =$

¹¹⁷ $\sum_{i=1}^{n} X_i/n$ towards a single number. However, in calculus, convergence of a sequence

¹¹⁸ $x_1, x_2, \dots$ is a strong assumption, and, typically, not even the (much weaker) Cesaro-

¹¹⁹ limit $\lim_{n \to \infty} \bar{x}_n = \lim_{n \to \infty}(\sum x_i/n)$ exists. In dynamic system theory, also, convergence

¹²⁰ towards a point is a rare exception.

¹²¹ In probability theory, the iid model represents a single population and a large, poten-

¹²² tially infinite sample from this population. To *avoid* convergence, it is thus straightfor-

¹²³ ward to consider *two* populations (distributions), say, $H_0$ and $H_1$, and a sample that

¹²⁴ fluctuates between them. In other words, if one switched between the populations skil-

¹²⁵ fully, $\bar{X}_n$ should not converge. In the jargon of dynamic system theory, the (unique)

¹²⁶ limit may be replaced by a (more complicated) attractor.

¹²⁷ However, a constant switching rate won't do: If $j$ observations from $H_0$ are followed

¹²⁸ by $j$ observations from $H_1$, and so forth, the arithmetic mean of this sequence will

¹²⁹ converge, since the 'influence' of another $j$ observations on $\bar{X}_n$ becomes insignificant

¹³⁰ with increasing $n$. Yet if $2^j$ observations from $H_0$ are followed by $2^{j+1}$ observations from



$H_1$, etc., one then obtains the desired effect. (On a logarithmic scale, taking $\mathrm{ld} = \log_2$, the ratio $\mathrm{ld}((2^{j+1})/2^j) = j + 1 - j = 1$ is a constant. Thus, there, one switches at a constant rate, '1' indicating that $H_0$ alternates with $H_1$.) Since $2^0 + 2^2 + 2^4 + \ldots$ observations are from $H_0$, and $2^1 + 2^3 + 2^5 + \ldots$ observations are from $H_1$, given a sample of size $2^n - 1$, considerably more than one half of these observations come from $H_0(H_1)$, if $n$ is an odd (even) number. Thus the arithmetic mean cannot 'settle' in some point.

Altogether, we obtain a stochastic process that is inhomogeneous in a particular way. Its paths depend on the concrete distributions of $H_0$ and $H_1$, and on the way switching is done. The aim of this article is to explore straightforward consequences of this setting.

# 3    The weaver's distribution

In order to keep things finite, suppose for the rest of this contribution that first moments exist, such that without real loss of generality $\mu(H_0) = 0$ and $\mu(H_1) = 1$ are the expected values of the two distributions involved.

A particularly simple way to alternate between $H_0$ and $H_1$ is to take the next batch of $2^j$ observations ($j = 0, 1, \ldots$) from population $H_0$ with probability $1 - p$, and from population $H_1$ with probability $p$. To avoid trivialities, we assume $0 < p < 1$ throughout this contribution. Thus, one creates a hierarchical random system (a particular random probability measure) composed of a choice mechanism which selects the population in charge, and a realization mechanism which provides observations from the population selected.

**Definition 1.** *(Exponential sampling)*

*Given two distributions $H_0$ and $H_1$, define exponential sampling as follows: A sample of size $2^n - 1$, i.e., $X_1; X_2, X_3; X_4, X_5, X_6, X_7; \ldots; X_{2^{n-1}}, \ldots, X_{2^n-1}$, consists of $n$ sub-samples, where the ith sub-sample $X_{2^{i-1}}, \ldots, X_{2^i-1}$ has size $2^{i-1}$ for $i = 1, \ldots, n$.*

*The selection mechanism $B$ chooses $H_1$ with probability $p$, and $H_0$ with probability $1 - p$ (independent of anything else, $0 < p < 1$). Thus, with these probabilities, the ith sub-*





*sample comes from $H_1$ or $H_0$, respectively. Finally, denote by $B_n$ the collection of $n$*

*such independent choices.*

With probability $p$, the first observation comes from $H_1$, and with probability $1 - p$,

the first observation comes from $H_0$. Thus, conditional on this choice, the expected

value observed is either $\mu(H_1) = 1$ or $\mu(H_0) = 0$, and the unconditional mean is

$\mu = p\mu(H_1) + (1 - p)\mu(H_0) = p.$

With probability $p$, the second *and* third observations both come from $H_1$, and with

probability $1 - p$, these observations both come from $H_0$. Thus, after two choices, the

overall situation is as follows:

| Number of observations from $H_0$ | Number of observations from $H_1$ | Probability | Conditional Mean |
|---|---|---|---|
| 1+2 | 0 | $(1 - p)^2$ | 0 |
| 1 | 2 | (1-p) p | 2/3 |
| 2 | 1 | p (1-p) | 1/3 |
| 0 | 3 | $p^2$ | 1 |

The unconditional mean does not change, since

$$\mu = p^2 + \frac{1}{3}p(1 - p) + \frac{2}{3}(1 - p)p = p^2 + p(1 - p) = p.$$

Similar to the binomial distribution, every path splits in two. However, unlike the

binomial distribution, the paths do not combine. Rather, like threads, they interweave.

**Illustration:** Binomial structure, local splitting (cascade), and global weaving

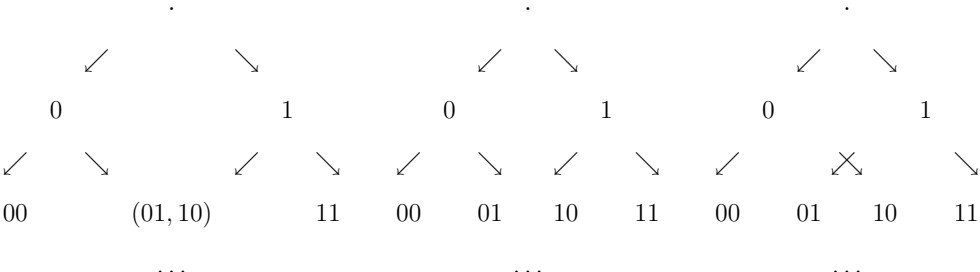





In a sense, the difference between splitting and weaving is minor: Given a binary string, the former operation adds the next cipher to the right (a suffix), whereas the latter operation adds the next cipher to the left (a prefix).

After $n$ steps (selections, choices), one thus obtains an interesting distribution:

**Theorem 2.** *(The weaver's distribution)*

*Given the situation described in Definition 1, suppose the first moments are $\mu(H_0) = 0$ and $\mu(H_1) = 1$, respectively.*

*For $n = 1, 2, \ldots$ let $S_n = \sum_{i=1}^{2^n-1} X_i$, $\bar{X}_n = S_n/(2^n - 1)$, and $Y_n = E(\bar{X}_n | B_n)$. Some elementary properties of these processes are:*

*(i) $Y_n$ assumes the values $y_k = y_{k,n} = k/(2^n - 1)$ for $k = 0, 1, \ldots, 2^n - 1$, and the difference between the realizations of $Y_n$ is a constant; more precisely,*

$$y_{k+1} - y_k = \frac{k+1}{2^n - 1} - \frac{k}{2^n - 1} = 1/(2^n - 1) \text{ for } k = 0, \ldots, 2^n - 2$$

*(ii) Suppose $B_n = \mathbf{b}_n$, then $\mathbf{b}_n = (b_{n-1}, \ldots, b_1, b_0)$ is a binary vector of length $n$, i.e., $b_{i-1} = 0$ if in the $i$th selection, $B$ chooses $H_0$, and $b_{i-1} = 1$ otherwise. Note that $b_{i-1}$ can also be interpreted as the $i$th digit in the binary representation of a natural number $k \in \{0, \ldots, 2^n - 1\}$, i.e., $k = \sum_{i=0}^{n-1} b_i 2^i$. Then the probability $p_k$ at the point $y_k$ is given by*

$$p_k = p^{\#1}(1-p)^{\#0} = p^{\sum_{i=0}^{n-1} b_i}(1-p)^{n-\sum_{i=0}^{n-1} b_i} \geq 0,$$

*where $\#1$ and $\#0$ denote the number of ones and zeros in $\mathbf{b}_n$, respectively. In particular, every $p_k$ can be written in the form $p_k = p^j(1-p)^{n-j}$ with some $j \in \{0, \ldots, n\}$.*



*(iii) More generally and explicitly, the distributions of $B_n$, $E(S_n|B_n)$, and $Y_n$ are*

| $(k)_{10}$ | $(k)_2$ | $\mathbf{b}_n$ | $E(S_n|\mathbf{b}_n)$ | $y_{k,n}$ | $p_k$ |
|---:|---:|---:|:---:|:---:|:---:|
| 0 | 0 | $(0,\dots,0)$ | 0 | 0 | $(1-p)^n$ |
| 1 | 1 | $(0,\dots,0,1)$ | 1 | $1/(2^n-1)$ | $p(1-p)^{n-1}$ |
| 2 | 10 | $(0,\dots,0,1,0)$ | 2 | $2/(2^n-1)$ | $p(1-p)^{n-1}$ |
| 3 | 11 | $(0,\dots,0,1,1)$ | 3 | $3/(2^n-1)$ | $p^2(1-p)^{n-2}$ |
| 4 | 100 | $(0,\dots,0,1,0,0)$ | 4 | $4/(2^n-1)$ | $p(1-p)^{n-1}$ |
| $\dots$ | $\dots$ | $\dots$ | $\dots$ | $\dots$ | $\dots$ |
| $2^n-5$ | $1\dots1011$ | $(1,\dots,1,0,1,1)$ | $2^n-5$ | $(2^n-5)/(2^n-1)$ | $p^{n-1}(1-p)$ |
| $2^n-4$ | $1\dots100$ | $(1,\dots,1,0,0)$ | $2^n-4$ | $(2^n-4)/(2^n-1)$ | $p^{n-2}(1-p)^2$ |
| $2^n-3$ | $1\dots101$ | $(1,\dots,1,0,1)$ | $2^n-3$ | $(2^n-3)/(2^n-1)$ | $p^{n-1}(1-p)$ |
| $2^n-2$ | $1\dots10$ | $(1,\dots,1,0)$ | $2^n-2$ | $(2^n-2)/(2^n-1)$ | $p^{n-1}(1-p)$ |
| $2^n-1$ | $\underbrace{1\dots1}_{n\ times}$ | $(1,\dots,1)$ | $2^n-1$ | 1 | $p^n$ |

Proof: (i) is obvious since $E(S_n|B_n)$ assumes the values $0,1,\dots,2^n-1$, and (ii) follows from (iii). (iii) holds by construction, or since by the binomial theorem $\sum_{k=0}^{2^n-1} p_k = \sum_{j=0}^{n}\binom{n}{j}p^j(1-p)^{n-j} = 1$. $\diamondsuit$

We say that $Y_n$ has a *weaver's distribution*, $Y_n \sim W(n,p)$, with parameters $n$ and $p$. Since powers of two play a major role, 'binary distribution' would also be a suitable choice - much in line with 'Bernoulli' and 'binomial' distributions, which are closely related.

**Theorem 3.** *(The geometric triangle)*

*Given the assumptions and the notation of the last theorem, let $\mathbf{b}_n = s_{ij}$ be a vector with exactly $i$ ones and $j$ zeros, such that $i + j = n$. Moreover, set $f = p/(1-p)$.*





(i) The probabilities $p(\cdot)$ of the concatenated vectors $(s_{ij}, 1), (1, s_{ij}), (s_{ij}, 0),$ and $(0, s_{ij})$

    are:

$$\frac{p(s_{ij}, 1)}{p(s_{ij}, 0)} = \frac{p(1, s_{ij})}{p(0, s_{ij})} = \frac{p^{i+1}(1-p)^j}{p^i(1-p)^{j+1}} = \frac{p}{1-p} = f$$

    In particular, $p_{k+1}/p_k = p/(1-p) = f$ for any two adjacent realizations $y_k, y_{k+1}$,

    and $k = 0, 2, \ldots, 2^n - 2$. The probabilities $p(\cdot)$ of the concatenated vectors $(0, 1, s_{ij}), (1, 0, s_{ij}),$

    etc., are

$$\frac{p(0, 1, s_{ij})}{p(1, 0, s_{ij})} = \frac{p(0, s_{ij}, 1)}{p(1, s_{ij}, 0)} = \frac{p(s_{ij}, 0, 1)}{p(s_{ij}, 1, 0)} = \frac{p^{i+1}(1-p)^{j+1}}{p^{i+1}(1-p)^{j+1}} = 1$$

(ii) For $n = 1, 2, \ldots,$ $p_0 = p_0(n) = (1-p)^n$ is the probability that only $H_0$ is chosen,

    and $p_k = p_0 \cdot f^{\#1}$ for $k = 0, \ldots, 2^n - 1$, where, again, $\#1$ is the number of

    ones in the binary representation of $k$. This means that the vector of probabilities

    $\mathbf{p}_n = (p_0, p_1, \ldots, p_{2^n-1})$ can be written as follows:

$$\begin{aligned}
\mathbf{p}_n &= p_0 \cdot (1; f; f, f^2; f, f^2, f^2, f^3; f, f^2, f^2, f^3, f^2, f^3, f^3, f^4; \ldots; \\
&\quad f, f^2, f^2, f^3, \ldots, f^{n-2}, f^{n-1}, f^{n-1}, f^n) = p_0 \cdot \mathbf{f}_n
\end{aligned}$$

(iii) More explicitly, with $\mathbf{p}_0 = 1$, the vector $\mathbf{f}_n$ has dimension $2^n$ and obeys the recursive relation $\mathbf{f}_0 = 1$, and $\mathbf{f}_n = (\mathbf{f}_{n-1}, f \cdot \mathbf{f}_{n-1})$ for $n = 1, 2, \ldots$ Thus its components can be calculated with the help of the following scheme, which may be interpreted as a geometric version of Pascal's triangle.[1]

$$
\begin{array}{llllllllllllllll}
n = 0: & & & & & & & & 1 & & & & & & & \\
n = 1: & & & & & 1 & & & | & & & f & & & & \\
n = 2: & & & 1 & & | & & f & & \| & & f & & | & & f^2 \\
n = 3: & 1 & | & f & \| & f & | & f^2 & \|| & f & | & f^2 & \| & f^2 & | & f^3
\end{array}
$$

$$\cdots$$

---

[1] Pascal named his triangle "triangle arithmetique." Thus, at least in French, it is straightforward to name the above multiplicative structure "triangle geometrique." Since row $n$ has $2^n$ entries, the geometric triangle is a 'real' triangle on the ld scale.





*Every row has $2^n$ entries. Note that the left and the right of every $|$ are 'separated'*

*by the factor $f$ in the following sense: First $[||]$, $1/f = f/f^2 = f^2/f^3 = \ldots$,*

*or, equivalently, $1 \cdot f = f; f \cdot f = f^2; f^2 \cdot f = f^3$, etc. Second $[|||]$, $(1, f) \cdot f =$*

*$(f, f^2); (f, f^2) \cdot f = (f^2, f^3), (f^2, f^3) \cdot f = (f^3, f^4)$, etc. Third $[||||]$, $(1, f, f, f^2) \cdot f =$*

*$(f, f^2, f^2, f^3); (f, f^2, f^2, f^3) \cdot f = (f^2, f^3, f^3, f^4)$; etc.*

*(iv) One may construct successive rows of (iii) in a rather elementary way: Start with*

*a single 1 in the very first row. Then, fork every entry of row $n$ into two, by*

*multiplying each entry with 1 and $f$ upon moving left and right, respectively. It is*

*quite remarkable that this local (cascade) view is equivalent to the global (weaving)*

*view taken in the definition.[2]*

   *(v) Applying the logarithm base $f$ to every entry of the geometric triangle yields the*

     *exponents, i.e., the following numbers:*

| $n$ | | | | | | | | | | | | | | | | | Sum $s_n$ |
|---|---|---|---|---|---|---|---|---|---|---|---|---|---|---|---|---|---|
| 0 | | | | | | | | 0 | | | | | | | | | 0 |
| 1 | | | | | 0 | | | $\|$ | | | 1 | | | | | | 1 |
| 2 | | | 0 | $\|$ | | 1 | $\|\|$ | | 1 | $\|$ | | 2 | | | | | 4 |
| 3 | 0 | $\|$ | 1 | $\|\|$ | 1 | $\|$ | 2 | $\|\|\|$ | 1 | $\|$ | 2 | $\|\|$ | 2 | $\|$ | 3 | | 12 |

          . . .

*In general, $s_0 = 0$, and $s_{n+1} = 2s_n + 2^n$ for $n = 0, 1, \ldots$ That is, one obtains the*

*sequence $0, 1, 4, 12, 32, 80, 192, 448, 1024, 2304, \ldots$*

Proof: (i) is proven in the statement of the theorem. However, (i) is also obvious, since

the positions of the numbers 0 and 1 are irrelevant for the probabilities in question. In

particular, for $k = 0, 2, \ldots, 2^n - 2$, the binary representations of $k$ and $k + 1$ differ in

exactly one position.

(ii) Using Theorem 2 (ii), one obtains immediately

---

[2]It may be noted that the 'weaver' is similar to the 'baker' in dynamic system theory. In particular, in both cases a locally defined transformation is closely related to global patterns. Theorem 10 connects the stochastic and the dynamic points of view explicitly.





$p_k = p^{\#1}(1-p)^{\#0} = p^{\#1}(1-p)^{n-(\#1)} = (1-p)^n \frac{p^{\#1}}{(1-p)^{\#1}} = p_0 f^{\#1}$

(iii) is a consequence of self-similarity. Since the binary representations of 0 and $2^{n-1}$,

and of 1 and $2^{n-1}+1$, etc., differ only by a single one,

$$
\begin{aligned}
\mathbf{p}_n &= (p_0, \ldots, p_{2^{n-1}-1}; p_{2^{n-1}}, \ldots, p_{2^n-1}) = (p_0, \ldots, p_{2^{n-1}-1}; fp_0, fp_1, \ldots, fp_{2^{n-1}-1}) \\
&= (\mathbf{p}_{n-1}, f\mathbf{p}_{n-1}) = (p_0 \mathbf{f}_{n-1}, fp_0 \mathbf{f}_{n-1}) = p_0(\mathbf{f}_{n-1}, f\mathbf{f}_{n-1})
\end{aligned}
$$

Since, again by (ii), also $\mathbf{p}_n = p_0 \mathbf{f}_n$, the desired result follows.

One may also prove (iii) by induction on $n$: First, $p_1 = fp_0$, and thus $(p_0, p_1) =$

$(p_0, fp_0) = p_0(1, f)$. Second, the binary representation of any $k \in \{0, \ldots, 2^n - 1\}$ is

a vector $\mathbf{b}_n = (b_{n-1}, \ldots, b_0)$. Let $\#1$ be the number of ones in $\mathbf{b}_n$. With probability

$1-p$, the next selection leads to $(0, \mathbf{b}_n)$, and with probability $p$ this selection results in

$(1, \mathbf{b}_n)$. Since in the first case, the number of ones does not change, and in the second

case, the number of ones increases by one, we obtain on the one hand (to the left),

$p_{i,n+1} = p_{0,n+1} f^{\#1} = (1-p)^{n+1} f^{\#1} = (1-p)p_{0,n} f^{\#1} = (1-p)p_{i,n}$ for $0 \le i \le 2^n - 1$. This

is tantamount to $\mathbf{f}_n$ being reproduced as the first half of $\mathbf{f}_{n+1}$. (Upon moving from $n$ to

$n+1$, the exponent of $f$ does not change.) On the other hand (to the right), we obtain

$p_{i,n+1} = p_{0,n+1} f^{(\#1)+1} = (1-p)^{n+1} f^{\#1} p/(1-p) = p(1-p)^n f^{\#1} = pp_{0,n} f^{\#1} = pp_{i,n}$ for

$2^n \le i \le 2^{n+1} - 1$. The additional factor $f$ means that the second half of $\mathbf{f}_{n+1}$ has to

be $f \cdot \mathbf{f}_n$.

(iv) The proof is by induction on $n$. For $n = 0$ there is nothing to prove, and the

equivalence is obvious for $n = 1$. By the inductive assumption, the vector occurring on

line $n$, having length $2^n$, has the form $\mathbf{w}_n = (\mathbf{l}_{n-1}, \mathbf{r}_{n-1}) = (\mathbf{l}_{n-1}, f \cdot \mathbf{l}_{n-1})$ where $\mathbf{l}_{n-1}$ is

a vector of length $2^{n-1}$. In other words, $r_k/l_k = f$ for $k = 1, \ldots, 2^{n-1}$.

Local splits (see the definition given in the statement of the theorem) produce a vector

$\mathbf{w}_{n+1}$ of length $2^{n+1}$. Since, locally, a step to the left reproduces the numbers, and a

step to the right multiplies any two entries on tier $n$ with the same factor $f$, we also

have, because of the inductive assumption, $w_{2^n+k}/w_k = f$ for $k = 1, \ldots, 2^n$. Therefore

$\mathbf{w}_{n+1} = (\mathbf{l}_n, f \cdot \mathbf{l}_n)$.



(v) Straightforward induction on $n$ yields the recursive formula. $\diamondsuit$

Note that the multiplicative triangle lies at the heart of the observation that "the best

known multifractal constructions use multiplicative operations" (Mandelbrot (1999),

p. 32).

**Theorem 4.** *(Further properties of the weaver's distribution).*

*Given the assumptions and the notation of Theorem 2, one obtains*

(i) *The probabilities corresponding to row $n$ can be constructed by the following simple*
*scheme:*

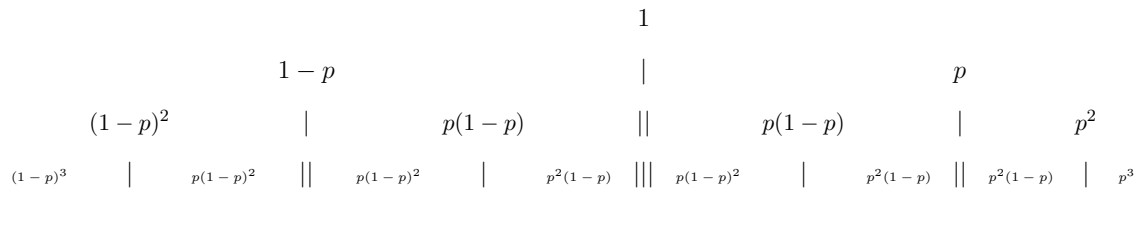

*Global interpretation [weaving]:* $\mathbf{p}_{n+1} = ((1-p)\mathbf{p}_n, p\mathbf{p}_n)$. *Local interpretation*

*[Bernoulli cascade]: Start with mass 1 in the very first (the zeroth) row. Then,*

*fork every probability of row $n$ into two, by multiplying each entry with $1-p$ (on*

*the left) and $p$ (on the right), respectively.*

(ii) *For $p > 1/2$, the sequence $p_0, fp_0, f^2p_0, \ldots$ increases. Accordingly, for $p < 1/2$,*

*we have $f < 1$. Therefore the sequence $p_0, fp_0, f^2p_0, \ldots$ decreases. If $p = 1/2$,*

*all probabilities coincide, i.e. we obtain the discrete uniform distribution on the*

*values $y_k = k/(2^n - 1)$; $p_k = 1/2^n$ for $k = 0, 1, \ldots, 2^n - 1$.*

(iii) *If $p > 1/2$, the mode occurs in one, and the median is larger than $1/2$. Vice versa,*

*if $p < 1/2$, the mode occurs in zero, and the median is less than $1/2$.*

(iv) *Symmetry: Suppose $Y \sim W(n, p)$ and $Y' \sim W(n, 1-p)$. Then $P(Y = y_k) =*

*$P(Y' = y_{2^n-1-k})$ for $k = 0, \ldots, 2^n - 1$.*



(v) *Distribution function $F$ of $W(n,p)$: For all $n \geq 0$ and $k = 0, \ldots, 2^n$ define $v_{k,n} = k/2^n$. For every fixed $n$, the mass left and right of $v_{k,n}$ $(0 < k < 2^n)$ is constant for every $m \geq n$, and so is the value of $F(v_{k,n})$. In particular, $F(v_{1,1}) = F(1/2) = (1-p)$ for all $n \geq 1$; $F(v_{1,2}) = F(1/4) = (1-p)^2$, $F(v_{3,2}) = F(3/4) = 1 - p^2$ for all $n \geq 2$; $F(v_{1,3}) = F(1/8) = (1-p)^3$; $F(v_{3,3}) = F(3/8) = (1-p)^2 + p(1-p)^2$, $F(v_{5,3}) = F(5/8) = (1-p) + p(1-p)^2$, $F(v_{7,3}) = F(7/8) = 1 - p^3$ for all $n \geq 3$, etc.*

(vi) *The total mass $p_k$ in every interval $[v_{k,n}, v_{k+1,n}]$ $(k = 0, \ldots, 2^n - 1)$ remains the same for all $m \geq n$. For $m = n$ it is located at the point $y_k = y_{k,n} = k/(2^n - 1)$. In the interest of consistency let $y_{0,0} = p$ and $p_0 = 1$ if $n = 0$.*

*Thus $W(n,p)$ may be interpreted as a discretisation of the density in the corresponding classical $p$ model.*

(vii) *Distribution of the jumps (stick heights): $F_n$ has $2^n$ points of discontinuity. If $p = 1/2$ there is a constant jump height $h = 1/2^n$. Otherwise, there are $n + 1$ different jump sizes, given by $h_j = p^j (1-p)^{n-j}$ for $j = 0, \ldots, n$, having a binomial distribution. That is, there is 1 jump of size $h_0 = (1-p)^n$, there are $\binom{n}{1} = n$ jumps of size $h_1 = (1-p)^{n-1} p$, etc.*

Proof:

(i) For $n = 1, 2, \ldots$, we have $p_0 = p_0(n) = (1-p)^n$ for the leftmost probability (only $H_0$ is selected). Applying the geometric triangle yields the result.

(ii) We have $p < 1/2 \Rightarrow f > 1$. Thus the mass in $y_1$ exceeds the mass in $y_0 = 0$ by the factor $f$, and the result follows straightforwardly.

(iii) is due to self-similarity. The claim for the mode can also be shown directly, since, if $p < 1/2$, we have $(1-p)^n < (1-p)^{n-k} p^k < p^n$.

(iv) Exchanging the roles of zeros and ones, and replacing $p$ by $1 - p$ yields the same distribution. In other words: The reflection of $W(n,p)$ across the axis of symmetry $y = 1/2$ is $W(n, 1-p)$.





(v) follows immediately from the geometric triangle. Geometrically speaking, the unit interval on the horizontal axis is successively halved. At the same time, the unit interval on the vertical axis is successively divided according to the ratio $f$. Thus, for finite $n \geq 1$, one obtains a step function with $2^n$ jumps.

(vi) holds because of the local interpretation of the geometric triangle: Each split can be interpreted as distributing the mass $p_k$ in $y_k$ to the points $y_{2k,n+1}$ and $y_{2k+1,n+1}$ in that same interval. Graphically, the stick of height $p_k$ in $y_{k,n}$ is broken into two sticks of heights $(1-p)p_k$ and $p \cdot p_k$, located in $y_{2k,n+1}$ and $y_{2k+1,n+1}$, respectively.

(vii) is due to construction. $\Diamond$

**Remark:** In the last theorem, the probabilities in (i) are the same as those in the classical $p$ model (de Wijs 1951, 1953). Its 'multifractal interpretation' is due to Mandelbrot (see, in particular, Mandelbrot (1989), section 5). Note, however, that the weaver's distribution is discrete and based on exponential sampling. Thus it obtains $2^n$ values.

Also note that there are two kinds of scale: the first could be named 'discrete time,' i.e., the total number of observations $t = 2^n$, the second would be 'logarithmic time,' that is, the number of selections, $\mathrm{ld}\, 2^n = n$.

## 4  Moments

**Theorem 5.** *(Expected value). Let $Y_n \sim W(n,p)$. Then, for every $n \geq 1$, the expected value of $Y_n$ is $p$.*

Proof: Let $\mu = EY_n$. One may decompose $\mu$ into a sum of $n$ terms $t_0, \ldots, t_{n-1}$, where the index $j$ counts the number of zeros in the corresponding binary vector $\mathbf{b}_n = (b_{n-1}, \ldots, b_0)$, that is, $j = n - \sum_{i=0}^{n-1} b_i$. More precisely, $\mu = \sum_{j=0}^{n-1} t_j = \sum_{j=0}^{n-1} p_j \cdot y_{[j]}$ where $y_{[j]}$ is the sum of all realizations with corresponding probability mass $p_j$.

$j = 0$: There is only one vector of dimension $n$ without the entry zero, i.e., $\mathbf{b}_n = (1, \ldots, 1)$. The corresponding probability is $p^n$ and thus $t_0 = 1 \cdot p^n$



$j = 1$: We have to consider the sum of all realizations of $Y_n$ that occur with probability

$p_1 = p^{n-1}(1-p)$, i.e. all binary sequences of length $n$, having exactly one zero. Thus

$$
\begin{aligned}
y_{[1]} &= \left(2^n - 1 - 2^0 + 2^n - 1 - 2^1 + 2^n - 1 - 2^2 + \ldots + 2^n - 1 - 2^{n-1}\right)/(2^n - 1) \\
&= (n2^n - n - \sum_{i=0}^{n-1} 2^i)/(2^n - 1) = (n(2^n - 1) - (2^n - 1))/(2^n - 1) = n - 1
\end{aligned}
$$

More intuitively, the number $2^n - 1$ is represented by a vector of $n$ successive ones in

the binary system. In the last equation we are looking for all sequences of length $n$

with exactly one zero. There are exactly $n$ such sequences, with the zero placed in each

possible position. Thus their sum is $n(2^n - 1) - (2^n - 1) = (n - 1)(2^n - 1)$. Dividing

by $2^n - 1$ yields the result, and $t_1 = (n - 1)p^{n-1}(1 - p)$.

In general, there are $\binom{n}{j}$ ways to place exactly $j$ zeros in a binary string of length $n$.

Without the zeros, the sum of these sequences would be $\binom{n}{j}(2^n - 1)$. However, for every

'chain' of zeros we have to subtract $\sum_{i=0}^{n-1} 2^i = 2^n - 1$, and there are $\frac{j}{n} \cdot \binom{n}{j}$ such chains.

Thus

$$
y_{[j]} = \left(\binom{n}{j}(2^n - 1) - \frac{j}{n}\binom{n}{j}(2^n - 1)\right)/(2^n - 1) = \binom{n}{j} - \binom{n-1}{j-1} = \binom{n-1}{j},
$$

and therefore $t_j = \binom{n-1}{j}p^{n-j}(1-p)^j$.

Putting everything together with the help of the binomial theorem, we get:

$$
\begin{aligned}
\mu &= \sum_{j=0}^{n-1} t_j = p^n + \sum_{j=1}^{n-1} \binom{n-1}{j}p^{n-j}(1-p)^j = p^n + \sum_{j=0}^{n-1} \binom{n-1}{j}p^{n-j}(1-p)^j - p^n \\
&= p \sum_{j=0}^{n-1} \binom{n-1}{j}p^{(n-1)-j}(1-p)^j = p \quad \diamondsuit
\end{aligned}
$$

After the first step, the distribution of the conditional expected values is $B(p)$. For any

random variable $X$ with values in the unit interval, and $EX = p$, this distribution has

maximum variance $p(1 - p)$. Upon weaving, probability mass is successively concen-

trated within the unit interval, and thus variance decreases. On the other hand, every

bifurcation may increase the variance term.



Both effects combined could result in a (net) monotone decrease of variance up to a

certain point. For concrete values, see the table on p. 26. Moreover, there should be a

limit variance $\sigma^2 = cp(1-p)$ with $c < 1$.

**Theorem 6.** *(Variance). Let $Y_n \sim W(n,p)$. Then the variance of this r.v. is*

$$\sigma^2(Y_n) = \frac{\sum_{i=0}^{n-1} 2^{2i}}{(2^n - 1)^2} p(1-p) \tag{1}$$

Proof: If we interpret $k = \sum_{i=0}^{n-1} b_i$ as a binary number of length $n$, the $i + 1$th

step of the above selection scheme defines its $i$th digit (from the right to the left,

$i = 0, \ldots, n-1$). Since the digits are independent by construction, every step con-

tributes a certain amount to the overall variance, independent of all of the other steps.

This means that the total variance can be decomposed into $n$ parts $\sigma_0^2, \ldots, \sigma_{n-1}^2$ that

sum to the total variance. The variance contributed by the $i$th digit is the difference

between $(? \cdots ? 1 ? \cdots ?)$ and $(? \cdots ? 0 ? \cdots ?)$, where the question marks denote arbitrary

other binary digits (the same for both numbers).

As a typical example, consider the case $n = 3$. The first step introduces variance that

can be assessed by means of considering two adjacent realizations of $Y_3$, e.g., the values

$0 = (000)_2$ and $1/7 = (001)_2/(111)_2$. This results in

$$\sigma_0^2 = p \left( \frac{1}{7} - \frac{1}{7} p \right)^2 + (1-p) \left( 0 - \frac{1}{7} p \right)^2 = \frac{1}{49} p(1-p) = \left( \frac{1}{7} \right)^2 p(1-p)$$

By the same token, the variance produced by the second step can be measured by two

realizations that differ only in the second component of their binary representation,

e.g., the values $0 = (000)_2$ and $2/7 = (010)_2/(111)_2$. This gives

$$\sigma_1^2 = p \left( \frac{2}{7} - \frac{2}{7} p \right)^2 + (1-p) \left( 0 - \frac{2}{7} p \right)^2 = \frac{4}{49} p(1-p) = \left( \frac{2}{7} \right)^2 p(1-p)$$

Finally, since the variance produced by the last step (consisting of 4 bifurcations) is

the same for all their descendants, it suffices to consider just one of these forks, e.g.,





$(00)_2$ and the values $0 = (000)_2$ and $4/7 = (100)_2/(111)_2$. This leads to

$$\sigma_2^2 = p\left(\frac{4}{7} - \frac{4}{7}p\right)^2 + (1-p)\left(0 - \frac{4}{7}p\right)^2 = \frac{16}{49}p(1-p) = \left(\frac{4}{7}\right)^2 p(1-p)$$

Putting everything together, we obtain $\sigma^2(Y_3) = \sigma_0^2 + \sigma_1^2 + \sigma_2^2 = (1+4+16)p(1-p)/7^2$.

Therefore, in general,

$$\sigma_i^2 = p\left(\frac{2^i}{2^n - 1} - \frac{2^i}{2^n - 1}p\right)^2 + (1-p)\left(0 - \frac{2^i}{2^n - 1}p\right)^2 = (2^i)^2 p(1-p)/(2^n - 1)^2$$

gives $\sigma^2(Y_n) = \sum_{i=0}^{n-1} \sigma_i^2 = p(1-p)\sum_{i=0}^{n-1}(\frac{2^i}{2^n-1})^2$. $\Diamond$

Note that the numerator shows an additive analogue to factorials: For factorials, $n! =$

$(n-1)! \cdot n$ holds. For the numerator, we have $\sum_{i=0}^{n}(2^i)^2 = \sum_{i=0}^{n-1} 2^{2i} + 2^{2n}$.

**Corollary 7.** *$EY_n^2$ exists, and so do all higher moments $EY_n^j$ for $j \geq 1$.*

Proof: For fixed $n$, all realizations $y_k$ are in the unit interval. Thus $y_k \geq y_k^2 \geq y_k^3 \geq \ldots$,

with strict inequality if $0 < y_k < 1$. Therefore $0 < EY_n^i < EY_n^j$ if $i > j$. $\Diamond$

**Lemma 8.** *The limit of the variance term is $\frac{1}{3}p(1-p)$*

Proof: Considered as a function of $n$, $\sigma^2(Y_n)$ is monotonically decreasing. Since it is

also nonnegative, it is clearly convergent. Moreover, a straightforward induction on $n$

shows that $\sum_{i=0}^{n-1} 2^{2i} = (2^{2n} - 1)/3$, thus

$$\frac{\sigma^2(Y_n)}{p(1-p)} = \frac{\sum_{i=0}^{n-1} 2^{2i}}{(2^n - 1)^2} = \frac{2^{2n} - 1}{3(2^{2n} - 2^{n+1} + 1)} = \frac{(2^{2n} - 1)/2^{2n}}{3(2^{2n} - 2^{n+1} + 1)/2^{2n}},$$

which converges to $1/3$ if $n \to \infty$. $\Diamond$





## 5  Limit distribution

Since, due to Theorem 4, the distribution function $F_n$ is well-known for all values $v(k, n)$, it is easy to pass to the limit. The limit function $F$ obviously is a distribution function.

**Theorem 9.** *(The weaver's hem)*

*Let $Y$ be the limit of $(Y_n)$, defined by its distribution function $F = \lim_{n \to \infty} F_n$. For obvious reasons, the corresponding distribution, i.e., $Y \sim W(p)$, may be called the weaver's hem.*

*$F$ is continuous, and the moments are $EY = p$ and $\sigma^2(Y) = p(1-p)/3$. Except for the case $p = 1/2$, when the discrete uniform distribution becomes the continuous uniform distribution on the unit interval (and thus $F$ is the identity function there), $F$ has no density with respect to Lebesgue measure.*

Proof: Using the notation of Theorem 4, for fixed $n$, all mass is concentrated at the points $y_{k,n} = k/(2^n - 1)$, $(k = 0, \ldots, 2^n - 1)$, and the jump heights there (Theorem 4 (vii)) go to zero if $n \to \infty$. Thus $F$ has to be continuous.

Because of $EX = \int_0^1 (1 - G(x))dx$ for any distribution function $G$ on the unit interval, and $F_n \to F$, we also have $EY = p$ for the weaver's hem. An analogous argument for the second moment and Theorem 7 yields $\sigma^2(Y) = p(1-p)/3$.

Rather heuristically, if $p > 1/2$, consider the interval $[0, 1/2[$. The mass of $1 - p$ available there is shifted to the left. Thus the distribution function grows rapidly at first, but hardly grows near $1/2$. Now consider the interval $]1/2, 1]$. Because a mass of $p$ is available there and systematically shifted to the left, the distribution function grows rapidly near $1/2$, but very slowly near $1$. Thus the distribution function has a sharp point at $1/2$ and cannot be differentiated there. The same holds for all $v(k, n)$. Since the set of these points lies dense in the unit interval, there should be no density.

Formally, consider the interval $[v_{k,n}, v_{k+1,n}]$ about $y_k = y_{k,n}$. For fixed $n$, this interval has length $v_{k+1,n} - v_{k,n} = (k + 1 - k)/2^n = 1/2^n$. By Theorem 3 (ii), the density in the



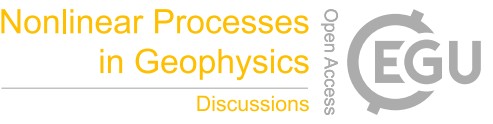

neighbourhood of $y_k$ is given by

$$g_{k,n} = 2^n p_k = 2^n p_0(n) f^{\#1} = 2^n (1-p)^n f^{\#1} = 2^n p^{\#1} (1-p)^{\#0}, \qquad (2)$$

where #0 and #1 are the number of zeros and ones in the binary representation of $k$,

respectively. If $p = 1/2$, $g_{k,n} = 1$, and thus $W(1/2)$ is the uniform distribution on $[0, 1]$.

In general, compare Equation (2) and the classical De Moivre-Laplace theorem. In the

latter case, one considers $\binom{n}{k} p^k (1-p)^{n-k}$, which approaches a limit $0 < c < \infty$, since

the convergence of $p^k (1-p)^{n-k}$ toward zero is counterbalanced by a sequence that goes

to infinity at the same speed, i.e., an appropriate binomial coefficient (also depending

on $n$ and $k$).

Here, every iteration ($n \to n+1$) doubles the number of values $y_k$, and thus the first

factor is $2^n$ instead of $\binom{n}{k}$. Moreover, due to Theorem 4, every $y_{k,n}$ is the starting

point of a cascade, i.e., a sequence of local bifurcations in the corresponding interval

$[v_{k,n}; v_{k+1,n}]$. After one iteration, the probabilities at $y_{2k,n+1}$ and $y_{2k+1,n+1}$, i.e., $(1-p)p_k$

and $p \cdot p_k$, respectively, differ by the factor $f$. After $l$ iterations, the probabilities at

the leftmost value $y_{2^l k, n+l}$ and the rightmost value $y_{2^l k + (2^l - 1), n+l}$ differ by $f^l$. If w.l.o.g.

mass is systematically shifted to the right ($p > 1/2$), we have $f > 1$, and thus the

ratio of these probabilities soon exceeds any bound. Even more so, $2^l (1-p)^l p_k \to 0$

and $2^l p^l p_k \to \infty$ in every interval $[v_{k,n}; v_{k+1,n}]$ if $l \to \infty$. Thus, there cannot be a limit

density. $\diamondsuit$

**Remarks:**

(i) Note that the 'roughness' of the density (measured by $f^l$) grows at the same rate

as the number of intervals. Thus $\ln f^n / \ln 2^n = \ln f / \ln 2$ is a constant, the fractal

dimension.

(ii) Studying the $p$ model, Riedi (1999) also builds on dyadic representations and

proves that the limit density does not exist (if $p \neq 1/2$). His first proof is similar

to ours, his second proof is based on the distribution function.





**Theorem 10.** *The weaver's hem and Mandelbrot's 'binomial measure' are equivalent.*

Proof: Mandelbrot's 'binomial measure' is the limit of the $p$ model, splitting the mass (locally) according to the geometric triangle. Thus, the $p$ model's Bernoulli cascade and weaving (see Theorem 4, (vi)) assign the same mass to every interval $[v_{k,n}; v_{k+1,n}]$. Since these intervals shrink to zero, the limit distributions have to coincide. $\diamondsuit$

# 6 The complete process

So far, we have mainly considered the distribution of the (conditional) expected values, $Y_n = E(\bar{X}_n | B)$, or, equivalently, the case of two one-point distributions located in $\mu(H_0)$ and $\mu(H_1)$, respectively. Looking at $\bar{X}_n$, however, there is not just variance between the populations $H_0$ and $H_1$, that we have considered so far, but also within each of these populations, $\sigma^2(H_0) = \sigma_0^2$ and $\sigma^2(H_1) = \sigma_1^2$, say, contributing to the total variance.

In complete generality, i.e., without specific distributional assumptions or any particular sampling scheme, let $n = n_0 + n_1$, and suppose that $n_0$ independent observations $Z_1, \ldots, Z_{n_0}$ come from the first population, and $n_1$ independent observations $Z'_1, \ldots, Z'_{n_1}$ come from the second population. At this point of sampling, the combined distribution is a mixture $M$ giving weight $n_0/n$ to the sample from $H_0$, and weight $n_1/n$ to the sample from $H_1$. In particular,

$$\bar{X}_n = \frac{\sum_{i=1}^n X_i}{n} = \frac{\sum_{i=1}^{n_0} Z_i + \sum_{i=1}^{n_1} Z'_i}{n} = \frac{n_0}{n} \frac{\sum_{i=1}^{n_0} Z_i}{n_0} + \frac{n_1}{n} \frac{\sum_{i=1}^{n_1} Z'_i}{n_1}$$

Thus we get the expected value (total mean)

$$\mu \quad = \quad E\bar{X}_n = E[E(\bar{X}_n | M)] = \frac{n_0}{n}\mu(H_0) + \frac{n_1}{n}\mu(H_1), \tag{3}$$



and variance

$$
\begin{aligned}
\sigma_n^2 &= \sigma^2(E(\bar{X}_n|M)) + E(\sigma^2(\bar{X}_n|M)) \\
&= \frac{n_0}{n}(\mu(H_0) - \mu)^2 + \frac{n_1}{n}(\mu(H_1) - \mu)^2 + \frac{n_0}{n}\frac{\sigma_0^2}{n_0} + \frac{n_1}{n}\frac{\sigma_1^2}{n_1}
\end{aligned}
\tag{4}
$$

**Theorem 11.** *(Expected value and variance)*

*With the assumptions of Theorem 2 , $E\bar{X}_n = p$ and*

$$
\sigma^2(\bar{X}_n) = p(1-p) + \frac{\sigma_0^2 + \sigma_1^2}{2^n - 1}
\tag{5}
$$

Proof: Sophisticated bookkeeping. Given exponential sampling, after $n$ selections, there are $2^n$ mixed distributions $Q_k$ ($k = 0, \ldots, 2^n - 1$) with the proportion $\lambda_k = k/(2^n - 1) = y_k$ of observations coming from $H_1$. In other words, $Q_k$ is a Bernoulli distribution $B(y_k)$. Distribution $Q_k$ occurs with probability $p_k$, where $p_k$ comes from a $W(n, p)$ distribution. If $Z_k \sim Q_k$, and $\mu_k = EZ_k$, Equation (3) translates into

$$
\mu = \sum_{k=0}^{2^n-1} p_k \mu_k = \sum_{j=0}^{n-1} p_j y_{[j]} = p
\tag{6}
$$

due to Theorem 5, using the notation of that theorem, that is, $y_{[j]}$ is the sum of all
$\mu_k = y_k = E(\bar{X}_n|B_n = (b_{n-1}, \ldots, b_0))$ with corresponding probability mass $p_j$. In
other words, the sum extends over all vectors $(b_{n-1}, \ldots, b_0)$ containing exactly $j$ zeros,
$j = n - \sum_{i=0}^{n-1} b_i$.

The first part of Equation (4), capturing the variance between the $Z_k$, reads

$$
\sigma^2(E(\bar{X}_n|B_n)) = \sum_{k=0}^{2^n-1} p_k(\mu_k - \mu)^2 = \frac{\sum_{i=0}^{n-1} 2^{2i}}{(2^n - 1)^2} p(1-p)
$$





due to Theorem 6. Finally, the second part of Equation (4), accounting for the variance

within the mixtures, becomes

$$E(\sigma^2(\bar{X}_n|B_n)) \;=\; \sum_{k=0}^{2^n-1} p_k \sigma^2(\bar{X}_n|B_n = (b_{n-1},\ldots,b_0)))$$

For every fixed $k = (b_{n-1},\ldots,b_0)_2$, $Q_k$ is a mixture with $k = \sum_{i=0}^{n-1} b_i 2^i$ observations

from $H_1$. Using $\mu(H_0) = 0$ and $\mu(H_1) = 1$, Equation (3) simplifies to $\mu_k = \lambda_k = $

$k/(2^n - 1)$ and the variance of $Z_k$, again according to Equation (4), is

$$
\begin{aligned}
\sigma^2(\bar{X}_n|B_n = (b_{n-1},\ldots,b_0)) &= (1-\lambda_k)\lambda_k^2 + \lambda_k(1-\lambda_k)^2 + (1-\lambda_k)\frac{\sigma_0^2}{2^n-1-k} + \lambda_k\frac{\sigma_1^2}{k} \\
&= (1-\lambda_k)\lambda_k + \frac{\sigma_0^2}{2^n-1} + \frac{\sigma_1^2}{2^n-1}
\end{aligned}
$$

Altogether we obtain the preliminary result

$$\sigma^2(\bar{X}_n) = \frac{\sum_{i=0}^{n-1} 2^{2i}}{(2^n-1)^2} p(1-p) + \sum_{k=0}^{2^n-1} p_k \left( \lambda_k(1-\lambda_k) + \frac{\sigma_0^2 + \sigma_1^2}{2^n-1} \right) \tag{7}$$

Now

$$
\begin{aligned}
\sum_{k=0}^{2^n-1} p_k \lambda_k(1-\lambda_k) &= \sum_{k=0}^{2^n-1} p^{\sum_{i=0}^{n-1} b_i} (1-p)^{n-\sum_{i=0}^{n-1} b_i} \frac{k}{2^n-1} \frac{2^n-1-k}{2^n-1} \\
&= \frac{1}{(2^n-1)^2} \sum_{k=1}^{2^n-2} p^{\sum_{i=0}^{n-1} b_i} (1-p)^{n-\sum_{i=0}^{n-1} b_i} \left( \sum_{i=0}^{n-1} b_i 2^i \right) \left( \sum_{i=0}^{n-1} (2^n-1-b_i 2^i) \right) \\
&= \frac{p(1-p)}{(2^n-1)^2} \Big\{ 1\cdot(2^n-2)(1-p)^{n-2} + 2\cdot(2^n-3)(1-p)^{n-2} \\
&\quad + 4\cdot(2^n-3)p(1-p)^{n-3} + \ldots + (2^n-3)\cdot 2\cdot p^{n-2} + (2^n-2)\cdot 1\cdot p^{n-2} \Big\}
\end{aligned}
$$





The last term in brackets can be rearranged:

$$
\begin{aligned}
\{\ldots\} &= (1-p)^{n-2}[1\cdot(2^n-2)+2\cdot(2^n-3)+4\cdot(2^n-5)+\ldots+2^{n-1}(2^{n-1}-1)] \\
&\quad +p(1-p)^{n-3}[3\cdot(2^n-4)+5\cdot(2^n-6)+\ldots+(2^{n-1}+2^{n-2})\cdot(2^n-1-(2^{n-1}+2^{n-2}))] \\
&\quad +\ldots+p^{n-2}[(2^{n-1}-1)2^{n-1}+\ldots+(2^n-5)\cdot4+(2^n-3)\cdot2+(2^n-2)\cdot1] \\
&= \binom{n-2}{0}(1-p)^{n-2}\left(\sum_{j=0}^{n-1}2^j(2^n-1-2^j)\right)+\binom{n-2}{1}p(1-p)^{n-3}\left(\sum_{j=0}^{n-1}2^j(2^n-1-2^j)\right) \\
&\quad +\ldots\binom{n-2}{n-3}p^{n-3}(1-p)\left(\sum_{j=0}^{n-1}2^j(2^n-1-2^j)\right)+\binom{n-2}{n-2}p^{n-2}\left(\sum_{j=0}^{n-1}2^j(2^n-1-2^j)\right) \\
&= \sum_{j=0}^{n-1}2^j(2^n-1-2^j)(1-p+p)^{n-2}=\sum_{j=0}^{n-1}2^j(2^n-1-2^j)
\end{aligned}
$$

so that

$$
\sum_{k=0}^{2^n-1}p_k\lambda_k(1-\lambda_k)=p(1-p)\sum_{j=0}^{n-1}2^j(2^n-1-2^j)/(2^n-1)^2
$$

and Equation (7) becomes

$$
\begin{aligned}
\sigma_n^2(\bar{X}_n) &= \frac{\sum_{i=0}^{n-1}2^{2i}}{(2^n-1)^2}p(1-p)+\sum_{k=0}^{2^n-1}p_k\lambda_k(1-\lambda_k)+\sum_{k=0}^{2^n-1}p_k\frac{\sigma_0^2+\sigma_1^2}{2^n-1} \\
&= \frac{\sum_{i=0}^{n-1}2^{2i}}{(2^n-1)^2}p(1-p)+\frac{\sum_{j=0}^{n-1}2^j(2^n-1-2^j)}{(2^n-1)^2}p(1-p)+\frac{\sigma_0^2+\sigma_1^2}{2^n-1} \qquad (8) \\
&= p(1-p)+\frac{\sigma_0^2+\sigma_1^2}{2^n-1},
\end{aligned}
$$

where the last equation is due to the next technical lemma. ◇

**Lemma 12.**

$$
\frac{\sum_{i=0}^{n-1}2^{2i}}{(2^n-1)^2}+\frac{\sum_{j=0}^{n-1}2^j(2^n-1-2^j)}{(2^n-1)^2}=1
$$

Proof: All one has to do is rearrange the terms:





$$\sum_{i=0}^{n-1} 2^{2i} \; + \; \sum_{j=0}^{n-1} 2^j(2^n - 1 - 2^j) = 2^0 + 2^2 + 2^4 + \ldots + 2^{2(n-1)}$$

$$+ \; 2^0(2^n - 1 - 2^0) + 2^1(2^n - 1 - 2^1) + 2^2(2^n - 1 - 2^2) + \ldots + 2^{n-1}(2^n - 1 - 2^{n-1})$$

$$= \; 2^0 + 2^0(2^n - 1 - 2^0) + 2^2 + 2^1(2^n - 1 - 2^1) + 2^4 + 2^2(2^n - 1 - 2^2)$$

$$+ \ldots + 2^{2(n-1)} + 2^{n-1}(2^n - 1 - 2^{n-1})$$

$$= \; (2^n - 1) + 2(2^n - 1) + \ldots + 2^{n-1}(2^n - 1)$$

$$= \; (2^n - 1)(1 + 2 + \ldots + 2^{n-1}) = (2^n - 1)(2^n - 1) \quad \diamond$$

It may be helpful to display some concrete values:

| | | $Denom.$ | $Weaving$ | $Mixing\ of\ H_0\ and\ H_1$ | $Proportions$ |
|---|---|---|---|---|---|
| $n$ | $2^n - 1$ | $(2^n - 1)^2$ | $\sum_{i=0}^{n-1}(2^i)^2$ | $\sum_{i=0}^{n-1} 2^i(2^n - 1 - 2^i)$ | $Weaving \leftrightarrow Mixing$ |
| 1 | 1 | 1 | 1 | 0 | $1 \leftrightarrow 0$ |
| 2 | 3 | 9 | 4 | 5 | $4/9 = 0.\bar{4} \leftrightarrow 0.\bar{5} = 5/9$ |
| 3 | 7 | 49 | 21 | 28 | $0.43 \leftrightarrow 0.57$ |
| 4 | 15 | 225 | 85 | 140 | $0.3\bar{7} \leftrightarrow 0.6\bar{2}$ |
| 5 | 31 | 931 | 341 | 620 | $0.35 \leftrightarrow 0.65$ |
| 6 | 63 | 3969 | 1365 | 2604 | $0.34 \leftrightarrow 0.66$ |

With respect to weaving and mixing, this means that one starts ($n = 1$) with a $B(p)$

distribution having expected value $p$ and variance $p(1-p)$ on the unit interval. In the

next steps, this 'available' variance is then distributed between weaving and mixing,

since due to Theorem 11 the latter variances add up to $p(1-p)$ for $n \geq 2$.

With $n$ increasing, Theorems 9 and 11 govern the asymptotic behaviour. That is, the

variance component due to weaving (i.e., the first term in Equation (8)) decreases

towards 1/3, which has the consequence that the component due to mixing (i.e., the

second term in Equation (8)) has to increase to 2/3. Moreover, since the variance within





the populations (i.e., the third term in Equation (8)) vanishes, we obtain the following

result:

**Theorem 13.** *(Limit distribution)*

*Given the assumptions and the notation of Theorem 2, if $H_0$ and $H_1$ both have finite*

*variances, then $B(p)$ is the limit distribution of the inhomogeneous (unconditional)*

*stochastic process $(\bar{X}_n)$.*

Proof: If there were just one population, $H_1$, say, $\bar{X}_n$ would converge to $\mu(H_1) = 1$

almost surely. Because $H_0$ makes $\bar{X}_n$ smaller, at least in expectation, 1 has to be

the largest cluster point. Since, for the same reasoning, 0 is the lowest cluster point,

$P(\bar{X}_n \notin [0,1]) \to 0$ if $n \to \infty$.

Now, because of the last theorem, for every $n$, the process is centered in $E\bar{X}_n = p$, and

its variance is given by Equation (5). Obviously, $(\sigma_0^2 + \sigma_1^2)/(2^n - 1) \to 0$. Thus we are

left with a limit distribution that is restricted to the unit interval, centered in $p$ and

has maximum variance $p(1-p)$. These properties imply the result. $\diamond$

Intuitively, the latter results are also quite obvious: If the variance within the popula-

tions vanishes, it is just the variance between the populations that is asymptotically

relevant. Since after all $H_0$ is selected with probability $1 - p$, and $H_1$ is selected with

probability $p$, the total variance is $p(1-p)$. One third of this variance is due to weaving

(i.e., the variance in the Weaver's hem), the remainder stems from mixing $H_0$ and $H_1$

(i.e., the mean variance of the mixtures $Q_k$). For finite $n$, weaving - or, equivalently, the

Bernoulli cascade - produces a $W(n, p)$ distribution with realizations $y_{k,n} = k/(2^n - 1)$,

$k = 0, \ldots, 2^n - 1$. Subsequently, every $y_{k,n}$ splits up into a $B(y_{k,n})$ distribution. Owing

to equation (8), their combined variance is $p(1-p)$ as well.

Of course, if the populations $H_0$ and $H_1$ are not too complicated, it is possible to study

the process $(\bar{X}_n)$ in much more detail.





## 7 Extensions

There are extensions on several tiers:

(i) Looking at Sections 2 and 3, it is straightforward to search for rather explicit formulas for higher moments, e.g. skewness or kurtosis of $W(n, p)$ and $W(p)$.

(ii) The binomial distribution is strongly connected with the arithmetic (Pascal's) triangle, and has a number of associated distributions: in particular, the normal, the multinomial, and the geometric distributions. Analogously, the weaver's distribution is strongly connected with a multiplicative structure (or the Binomial cascade), and apart from Mandelbrot's limit distribution, other distributions are associated with it. In particular, two generalizations of the geometric distribution are straightforward:

Suppose the process stops upon encountering the first one. If this occurs in step $i$, the classical geometric distribution takes the realization $i$ occurring with probability $(1-r)^{i-1}r$. Here, it is more natural to consider the value $2^{i-1}$. Suppose the random variable $T$ has such an 'extended' geometric distribution. Then

$$ET = r + 2r(1-r) + 4r(1-r)^2 + 8r(1-r)^3 + \ldots = r \sum_{i=0}^{\infty} 2^i(1-r)^i.$$

Since $\sum_{i=0}^{\infty} 2^i(1-r)^i = \sum_{i=0}^{\infty}(2-2r)^i$ is a geometric series that converges if its argument $2-2r$ is less than 1, convergence occurs if and only if $r > 1/2$. Moreover, the same kind of reasoning yields that

$$ET^2 = r + 4r(1-r) + 16r(1-r)^2 + 64r(1-r)^3 + \ldots = r \sum_{i=0}^{\infty} 4^i(1-r)^i$$

converges if $r > 3/4$.

Thus, altogether, there are three different kinds of behaviour:

a) If $r > 3/4$, then $ET$ and $\sigma^2(T)$ both exist.

b) If $1/2 < r \le 3/4$, $ET$ exists, but not $\sigma^2(T)$.





c) If $r \leq 1/2$, then neither the first nor the second moment of $T$ exists.

In a sense, it is also straightforward to take the realizations of the weaver's distribution, that is $y_i = 2^{i-1}/(2^i - 1)$ for $i = 1, 2, \dots$ This approach yields

$$ET' = r \sum_{i=1}^{\infty} \frac{2^{i-1}}{2^i - 1}(1 - r)^{i-1} \leq r \sum_{i=1}^{\infty} (1 - r)^{i-1} = r/r = 1.$$

Since the series is monotonically increasing, it converges for every $r > 0$.

(iii) Looking at Theorem 11, if the variance of the populations is infinite, the last term in (5) need not vanish asymptotically. Thus, in the limit, the variances of weaving and mixing are augmented by variance components stemming from within the populations. It would be interesting to know how this phenomenon changes the limit distribution. In particular, this approach offers a constructive way to deal with populations that have nonexistent second moments.

(iv) Other multiplicative schemes come to mind, in particular involving dependencies among the random variables, and with time-dependent $p_n$ (e.g., Mandelbrot (1974), Serinaldi (2010), Lovejoy and Schertzer (2013), Cheng (2014)). It would also be interesting to learn more about the relationship between local cascades and global weaving (or shuffling) in general.

(v) Pascal's triangle is additive, whereas the geometric triangle is multiplicative. An alternative view would be that splitting and merging alternate in Pascal triangle, whereas there are only splits in the geometric triangle, since the latter can be interpreted as a Bernoulli cascade. In general, the dual operations of splitting and merging could alternate in more complicated (deterministic or random) ways.

(vi) With respect to the two-population interpretation it is straightforward to alternate between $H_0$ and $H_1$: The first observation comes from $H_0$, then two observations come from $H_1$, another four observations come from $H_0$, etc. However, this deterministic way to proceed introduces an asymmetry, since it makes a difference which population comes first.





Moreover, the latter regime is a special case of the following, more general (and also more promising) Markov scheme. That is, given two populations and the present state, one switches according to the following transition matrix

|  | to: | $H_0$ | $H_1$ |
|---|---|---|---|
| from | $H_0$ | $s$ | $1-s$ |
|  | $H_1$ | $1-s$ | $s$ |

Here, $s = 1$ corresponds to the classical situation (all the observations come from one of the two populations), and $s = 0$ corresponds to deterministic switching at the highest frequency possible (always).

It is possible to characterize the behaviour of $\bar{X}_n$ qualitatively, and w.l.o.g let $\mu(H_0) = 0$ and $\mu(H_1) = 1$, respectively. On the one hand, if switching is rare, some $H_i$ is selected and the arithmetic mean of the sample is 'apparently converging' towards $\mu(H_i)$. These (long) phases of stability are interrupted by sudden switches (within a few time periods on the $\log t$ scale) to the other population. In other words, 0 and 1 are strongly 'attracting' $(\bar{X}_n)$. On the other hand, if switching is frequent, the process spends most of the time oscillating between $H_0$ and $H_1$, i.e., in a subset of the unit interval, and may even converge (just think of the sequence $0, 1, 0, 1, \ldots$). Thus there seem to be many possible limit distributions 'between' some constant $c \in [0, 1]$ and some Bernoulli $B(p)$.

Moreover, following the 'deterministic' track, it is straightforward to consider different (possibly time-dependent) switching rates. Thinking along probabilistic lines, asymmetric switching should be studied, that is, transition matrices

$$\begin{pmatrix} s & 1-s \\ 1-s' & s' \end{pmatrix} \quad \text{with} \quad s \neq s'.$$

(vii) If switching occurs very often, or if a constant switching regime (on the $\log t$-scale) is employed ($n$ observations from $H_0$, $n$ observations from $H_1$, etc.), we are back to the classical theory ($\bar{X}_n$ converging to a fixed number). If switching occurs seldom, in particular, if sampling is exponential, $\bar{X}_n$ converges in distribution. It would be





interesting to know more about the 'line' separating these two situations. What
are necessary and sufficient conditions for either kind of convergence of $\bar{X}_n$?

Seen a bit differently, calculus deals with convergent series, i.e., $(x_i)$ converges
by itself. Classical probability theory deals with the case of existing expected
value, i.e., the statistic $\sum X_i/n$ converges. Throughout this contribution, $\sum X_i/n$
fluctuates in a (stochastically) regular way, generating a simple fractal structure.

Therefore, further extensions seem very plausible: e.g., by considering other, more
complicated summation schemes (Volkov 2001), employing more complex switch-
ing regimes, a larger number of populations, or multidimensional distributions.
Higher levels of complexity may thus be reachable in a rather systematic manner.

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
