# Peer review of "In-depth analysis of a discrete p model"

_Nonlinear Processes in Geophysics, 2019_

## Referee Comment (RC1) · Anonymous Referee #1 · 15 Apr 2020

**REPORT ON**
**IN-DEPTH ANALYSIS OF A DISCRTE P MODEL**
**SUBMITTED TO**
**NONLINEAR PROCESSES IN GEOPHYSICS**

According to the author, the paper introduces a discrete version of $p$-model based on a new sampling. Let us recall the model. The author is interested in

$$S_n = \sum_{i=1}^{2^n-1} X_i, \quad \overline{X}_n = \frac{S_n}{2^n-1}, \quad Y_n = \mathbb{E}(\overline{X}_n | B_n)$$

where $X_1, X_2, \cdots, X_{2^n-1}$ represent the sampling, which is made as follows: the elements in the $j$-th group

$$X_{2^{j-1}}, X_{2^{j-1}+1}, \cdots, X_{2^j-1} \qquad (1 \le j \le n)$$

are all (at the same time) selected to be 1 with probability $p$ and 0 with probability $1-p$; where $B_n$ represents the state of selection. A realization of $B_n$ can be identified with a binary vector $(b_{n-1}, \cdots, b_1, b_0)$ ($b_{j-1} = 1$ meaning that 1 is selected for the $j$-th group), which can be identified with an integer betweet 0 and $2^n-1$ via the binary representation. The selections are assumed independent. Thus, let

$$Z_j = \sum_{i=2^{j-1}}^{2^j-1} X_i.$$

Then

$$S_n = \sum_{j=1}^{n} Z_j. \tag{1}$$

Observe that $Z_j$'s are independent and

$$P(Z_j = 2^{j-1}) = p, \quad P(Z_j = 0) = 1 - p. \tag{2}$$

It is then clear that $S_n$ is a variant of binomial variable and it is rather direct to derive its proprties like those stated in the paper. For example,

$$\mathbb{E}S_n = \sum_{j=1}^{n} \mathbb{E}Z_j = p \sum_{j=1}^{n} 2^{j-1} = p(2^n - 1).$$

Thus Theorem 5 stating that $\mathbb{E}Y_n = p$ is trivial, because the expectation of $Y_n$ is equal to that of $S_n$ divided by $2^n - 1$. Also notice that

$$Y_n = \frac{1}{2^n-1} \sum_{j=1}^{n} b_{j-1} 2^{j-1} \tag{3}$$

where $b_0, b_1, \cdots, b_{n-1}$ are independent and $p$-Bernoulli random variables. This allows us to quickly obtain Theorem 6. The representation (3) should be a key for obtaining other results, including that on the limit distribution of $Y_n$ (Theorem 13).

I don't understand why $H_0$ and $H_1$ are variables, and their expectations $\mu(H_0)$ and $\mu(H_1)$ and monents are mentioned.

Mathematically, things can be much simplfied and easily obtained. The study of the limit law of $Y_n$ has some interests.

It is a long paper with many discussions on different physical topics. But the relation between the studied model and these topics are not really discussed.

---

## Author Comment (AC1) · 24 Apr 2020

**Response to Reviewer #1**

First of all, I would like to thank this reviewer for his or her thoughtful comments and the potential simplifications he or she has pointed out.

A crucial statement can found towards the end of his / her report: "I don't understand why $H_0$ and $H_1$ are variables, and their expectations $\mu(H_0)$ and $\mu(H_1)$ and moments are mentioned." Therefore the referee thinks that, upon selection, $X_i$ is a constant (being either zero or one).

However, the author treats the general case of two populations with nontrivial distributions and different expected values $\mu(H_0) \neq \mu(H_1)$. The assumption that $\mu(H_m) = m, (m = 0, 1)$ is only made in order to simplify the formulas. In particular, $m$, the index of the population, should not be confused with some realization $X_i = x_i$.

Despite this misconception, it is instructive to understand the main argument of the referee, i.e., to look at the trivial example of degenerate (constant) populations. Moreover, since the random variables $X_i$ are organized in groups (subsamples), it is indeed helpful to consider $Z_j = \sum_{i=2^{j-1}}^{2^j-1} X_i$ $(j = 1, \ldots, n)$ and the decomposition $S_n = \sum_{j=1}^n Z_j = \sum_{j=1}^n \sum_{i=2^{j-1}}^{2^j-1} X_i$.

However, it is not true that $Z_j$ only assumes the values $2^{j-1}$ and 0 which would imply that $S_n \in \{0, \ldots, 2^n - 1\}$, and $S_n$ could be considered some "variant of binomial variable." Alas, since the $X_i$'s have arbitrary distributions, $S_n = s_n$ may be any real number.

Rather, due to exponential sampling, the Weaver's distribution $W(n, p)$ extends $B(n, p)$ considerably. Since the basic building block of both distributions is the Bernoulli $B(p)$, they are related, but there is also a marked difference (see Definition 1 and Theorems 2-4). In particular, the paths defining the Weaver only split and never merge (see the Illustration, p. 8), and the limit distribution $W(p)$ is a fractal instead of being the Normal.

Nevertheless, since the selection process $\mathbf{B} = (B_0, \ldots, B_{n-1})$ is a vector of independent Bernoullis, there is some "built-in" Binomial distribution. My proofs of Theorems 5, 6 and the first part of Theorem 11 rely on the construction process of $W(n, p)$, display the "built-in" cascade and/or weaving mechanism, and use the associated Binomial.

The referee is right that they can be simplified. The crucial observation is that the total variance in $S_n$ (or, equivalently, in $\bar{X}_n$) stems from the independent Bernoulli selections ($H_0$ or $H_1$) and the variance in the distributions $\mathcal{L}(X_i)$. Since the $Y_n$'s are *conditional expectations*, only the selection procedure is of relevance to them, and thus there is the elegant representation

$$(2^n - 1)y_{k,n} = E(S_n|(b_0, \ldots, b_{n-1})) = \sum_{j=1}^n b_{j-1} 2^{j-1} \in \{0, \ldots, 2^n - 1\},$$

mentioned in Theorem 2. This representation may be used in Theorem 5,

$$
\begin{aligned}
E(Y_n) &= E(\sum_{j=1}^n B_{j-1} 2^{j-1})/(2^n - 1)) = \frac{1}{(2^n - 1)} \sum_{j=1}^n E(B_{j-1} 2^{j-1}) \\
&= \frac{1}{(2^n - 1)} \sum_{j=1}^n 2^{j-1} E(B_{j-1}) = p,
\end{aligned}
$$

and in Theorem 6:

$$\sigma^2(Y_n) = \sigma^2(\sum_{j=1}^{n} B_{j-1} 2^{j-1})/(2^n - 1)) = \frac{1}{(2^n - 1)^2} \sum_{j=1}^{n} \sigma^2(B_{j-1} 2^{j-1})$$

$$= \frac{1}{(2^n - 1)^2} \sum_{j=1}^{n} (2^{j-1})^2 \sigma^2(B_{j-1}) = \frac{p(1-p)}{(2^n - 1)^2} \sum_{j=1}^{n} 2^{2(j-1)}$$

The expression $\sum_{j=1}^{n} 2^{2(j-1)}/(2^n - 1)^2$ has a nice geometric interpretation that is used (implicitly) in the proof of Lemma 12.

| | $1$ | $2$ | $4$ | $\ldots$ | $2^{n-1}$ | $\sum$ |
|---|---|---|---|---|---|---|
| $1$ | $\mathbf{1}$ | $2$ | $4$ | $\ldots$ | $2^{n-1}$ | $2^n - 1$ |
| $2$ | $2$ | $\mathbf{4}$ | $8$ | $\ldots$ | $2^n$ | $2(2^n - 1)$ |
| $4$ | $4$ | $8$ | $\mathbf{16}$ | $\ldots$ | $2^{n+1}$ | $4(2^n - 1)$ |
| $\ldots$ | $\ldots$ | $\ldots$ | $\ldots$ | $\ldots$ | $\ldots$ | $\ldots$ |
| $2^{n-1}$ | $2^{n-1}$ | $2^n$ | $2^{n+1}$ | $\ldots$ | $\mathbf{2^{2(n-1)}}$ | $(2^{n-1})(2^n - 1)$ |
| $\sum$ | $2^n - 1$ | $2(2^n - 1)$ | $4(2^n - 1)$ | $\ldots$ | $(2^{n-1})(2^n - 1)$ | $(2^n - 1)^2$ |

That is, $\sum_{j=1}^{n} 2^{2(j-1)}$ is the trace of the above matrix and $\sum_{j=1}^{n} 2^{2(j-1)}/(2^n - 1)^2$ is the proportion of the trace relative to the whole square.

For the first part of Theorem 11 note that $E\bar{X}_n = ES_n/(2^n - 1)$, and

$$ES_n = E\left(\sum_{j=1}^{n}\sum_{i=2^{j-1}}^{2^j - 1} X_i\right) = \sum_{j=1}^{n}\sum_{i=2^{j-1}}^{2^j - 1} EX_i = \sum_{j=1}^{n}\sum_{i=2^{j-1}}^{2^j - 1} p = (2^n - 1)p,$$

since $EX_i = p \cdot \mu(H_1) + (1 - p) \cdot \mu(H_0) = p$. (All random variables of some group $j$ have distribution $H_1$ with probability $p$, and $H_0$ with probability $1 - p$. Thus these are also the corresponding probabilities of any single $X_i$ $(i = 1, \ldots, 2^n - 1)$.)

The second part of Theorem 11 considers the variance $\sigma^2(\bar{X}_n)$. It turns out that it is crucial to study $Z_j$, which is the sum of $2^{j-1}$ random variables. Selection $j$ chooses $H_1$ with probability $p$, and $H_0$ with probability $1 - p$. Given $H_m$ $(m = 0, 1)$, the conditional variance of $Z_j$ is $\sigma^2(Z_j|H_m) = 2^{j-1}\sigma^2(X_i|H_m) = 2^{j-1}\sigma_m^2$, since the $X_i$'s are independent.

Moreover, $E(Z_j|H_m) = 2^{j-1}m$, and $EZ_j = 2^{j-1}p$. A variance decomposition of $Z_j$ yields

$$\sigma^2(Z_j) = p(E(Z_j|H_1) - EZ_j)^2 + (1 - p)(E(Z_j|H_0) - EZ_j)^2$$
$$+ p\sigma^2(Z_j|H_1) + (1 - p)\sigma^2(Z_j|H_0)$$
$$= p(2^{j-1} - 2^{j-1}p)^2 + (1 - p)(0 - 2^{j-1}p)^2 + p2^{j-1}\sigma_1^2 + (1 - p)2^{j-1}\sigma_0^2$$
$$= 2^{2(j-1)}p(1 - p)(1 - p + p) + 2^{j-1}p\sigma_1^2 + 2^{j-1}(1 - p)\sigma_0^2$$

which implies

$$
\begin{aligned}
\sigma^2(\bar{X}_n) &= \sigma^2\left(\frac{S_n}{2^n-1}\right) = \frac{1}{(2^n-1)^2}\sigma^2(S_n) = \frac{1}{(2^n-1)^2}\sum_{j=1}^{n}\sigma^2(Z_j) \\
&= \frac{\sum_{j=1}^{n}2^{2(j-1)}}{(2^n-1)^2}p(1-p) + \frac{\sum_{j=1}^{n}2^{j-1}}{(2^n-1)^2}p\sigma_1^2 + \frac{\sum_{j=1}^{n}2^{j-1}}{(2^n-1)^2}(1-p)\sigma_0^2 \\
&= \sigma^2(Y_n) + \frac{p\sigma_1^2 + (1-p)\sigma_2^2}{2^n-1}.
\end{aligned}
\tag{1}
$$

Consistently, the variance within the population washes out quickly, and one obtains $\lim_{n\to\infty}\sigma^2(\bar{X}_n) = \sigma^2(Y) = p(1-p)/3$. This means, that the limit distribution of $\bar{X}_n$ is the same as that of $Y_n$, i.e., $W(p)$. In other words, within the populations, the LLN applies. Therefore, in the limit, only the variance caused by exponential sampling (the selection process) is relevant.

Although equation (7) is wrong and should be replaced by the above equation (1), the idea of Theorem 13 is still valid, and thus that Theorem may also be "repaired:"

Starting with $Y_n \sim W(n,p)$, one may replace each realization $y_{k,n} = k/(2^n-1)$ by a Bernoulli r.v. $C_{k,n} \sim B(y_{n,k})$. The r.v. $C_{k,n} \circ Y_n$ assumes the values zero and one, since each $y_{k,n}$ is mapped to 1 w.p. $y_{k,n}$ and to 0 w.p. $1 - y_{k,n}$. Since the location of the distribution does not change if on splits the realizations $y_{k,n}$ in the aforementioned way, $E(C_{k,n} \circ Y_n) = EY_n = p$, and thus $C_{k,n} \circ Y_n \sim B(p)$. For fixed $n$, one might think of the collection of all $C_{k,n}$ ($k = 0, \ldots, 2^n - 1$) as a family of "dual distributions" to $W(k,n)$. In the paper, this train of thought is called "mixing," and it is still true that the total variance $p(1-p)$ is the sum of "weaving and mixing."

In a nutshell, owing to the "elementary" construction principles used throughout the manuscript, explicit formulas can be given for the crucial parameters of the processes and their limits. Because of Theorem 10, at least some of these results can be applied to the received situation. Moreover, Theorem 10 demonstrates that a multifractal discrete structure (i.e., $W(p)$) derived in an iterative "bottom up" way[1] may be equivalent to a structure obtained with a similar iterative "top down" procedure.[2] At least to the author, this looks like an exemplar of a more general sandwich principle.

Of course, to some extent, it is a matter of taste how many of the extensions should be discussed in Section 7.

Finally, the author agrees with the reviewer that the core of this contribution is a mathematical analysis of an important nonlinear model that is applied in many fields.
* * *
[1]starting with $B(p)$ or point mass at $p$, if one defines $W(k,0) = \varepsilon_p$

[2]starting with the uniform distribution on the unit interval

---

## Referee Comment (RC2) · Anonymous Referee #2 · 24 Jun 2020

**A review of "In-depth analysis of a discrete p model " by U. Saint-Mont**

**General comment**

The author largely motivates his paper by foreseen applications to cascade and multifractal processes, at least to the "*p*-model": "It is the aim of this contribution to introduce original concepts that shed new light on the latter paradigmatic cascade and allow key features to be derived in a rather elementary fashion". This goal is in agreement with the scope of NPG and the interests of its readership. But, this goal does not seem to be achieved in the present manuscript that furthermore introduces often complexity instead of claimed simplicity, e.g., introducing variables that are finally scarcely used. The algebra is often tedious, in particular demonstrations are not always as straightforward as they could be, definitions are not always precise and from time to time missing. The introduction of non obvious jargon terms does not help the reader who tries to decipher the present manuscript. For instance, there is a given uncertainty on which "*p*-model" is considered, whereas this model was claimed to be the main topic of this paper. In fact, it is not clear what is the added value of this paper with respect to earlier papers on multiplicative cascades: it seems to be concerned with much milder processes, sharing only superficial properties.

Overall, I consider that this manuscript is not publishable in its present form, but requires to be thoroughly clarified, including its goal; it thus requires a major revision that could be quite challenging.

**Detailed comments**

Which *p*-model?

In the introduction, the author presents the *p*-model as the iterative splitting "in proportion" $1 - p$ and $p$ respectively on the left and right subsegments of a (uniform) distribution over the initial segment ([0,1]). This is somewhat close to de Wijs (1951), who used the notations (1+*d*) and (1-*d*) for these proportions, where *d>0* is the "dispersion coefficient". In the later case, the model is "micro-canonical" because it strictly preserves in a deterministic manner the content at each cascade step (simply because: $(1 + d)/2 + (1 - d)/2 = 1$). Combining the two notations and respecting the positivity of the content "proportions" requires to identify *p* to 1+*d* (inverting left and right does not hurt!) and therefore *p>1*. One may note that the $\alpha$-model corresponds to a stochastic generalisation of this model, with only a canonical conservation of the content, i.e., only on the statistical average.

However, in the rest of the paper *p* becomes a probability, with therefore the requirement *p≤1*, and the values $\mu(H_0) = 0$ and $\mu(H_1) = 1$ ("without real loss of generality"), with probability 1-*p* and *p*. This rather corresponds to the $\beta$-model, in fact a special case of the $\alpha$-model, both being stochastic, canonical, multiplicative cascade models. The main difference is that the $\beta$-model is the exceptional case of mono-/uni- fractality, i.e., its support has a unique fractal dimension (in fact defined by $\beta$ =*p*), whereas other $\alpha$ -models are multifractal models, with possible divergence of statistical moments.

Unfortunately, the precise reference to the page 329 of Mandelbrot (1974), which could have helped to clarify what the author has in mind, is outside of the page range (331-358) of this paper.

Unnecessary developments.

With respect to applications to cascade models, all the developments around the so-called "exponential sampling" seem rather useless, as well as the variables $X_i$. Indeed, what could be the

.

foreseen advantage to introduce these variables which contain larger and larger numbers of identical replica of the same variable for increasing $i$? It is indeed much better to focus on the binary variable $b_j$ (=0,1) that could be compacted into vectors $\mathbf{b}_n = (b_0, \ldots b_j \ldots, b_{n-1})$ or even better into dyadic expansions $\sum_{j=0}^{n-1} b_j 2^{j-n}$. The latter is certainly the most interesting one, because these expansions have been often used for the "coordinates" of subsegments of the cascade $\sum_{j=0}^{n-1} x_j 2^{j-n}$. Surprisingly, these subsegments have been only evoked in the introduction.

Missing information.
Very surprisingly there is no clear indication on how to proceed from step n to the step n+1, contrary to what is explicitly done for multiplicative cascades. The reader is rather invited to infer it from an "illustration" (bottom p.8, without a reference number and axis labels) and a table of a few examples (p.9, again without a reference number). Furthermore, Definition 1 is rather ambiguous: it could be understood that at each step is drawn independently of the previous one. In this case, the components of $\mathbf{b}_n$ are merely n independent p-Bernoulli variables, which render trivial many announced properties (e.g., theorem 2) but the relation with a multiplicative cascade has to be done. Due to missing informations, it is often difficult to have a definitive opinion what is really demonstrated. This is particularly the case of theorem 10, which seems to only state that the $Y$ process and a $p$-model (still not perfectly defined) have a common type of probability distribution and therefore it does not shed any new light on the $p$-model.

Tedious algebra
In the framework of the previous hypothesis, not only the computations of the mean of $Y_n$ (theorem 5) and its variance $\sigma^2(Y_n)$ (theorem 6) are trivial, as already pointed out by referee 1, because $Y_n$ is then simply the sum mutually independent variables $b_j 2^j$, normalised by their number $2^n - 1$, but the resulting expression for $\sigma^2(Y_n)$ (Eq.1) can be further simplified, in particular to immediately obtain its asymptotic value for $n \to \infty$ (Lemma 8), without any recourse to induction. The same is true for the theorem 11 (including Lemma 12) that is particularly lengthy and awkward. In a general manner, NPG readers will not enjoy many algebraic developments that are too much elementary (e.g., on geometric series).

The "complete process"
All results of the first 5 sections were presented for the conditional variable $Y$, furthermore with $\mu(H_0) = 0$ and $\mu(H_1) = 1$ ("without real loss of generality"). The section 6 abruptly introduces some inner variability in the populations $H_0$ and $H_1$, and therefore considers the non-conditional variable $\bar{X}$. It seems that finite variances only introduce marginal fluctuations (theorem 11). This is interesting, although expected for a linear process. However, it should have been motivated, since it does not seem to help to be closer to a cascade process.

.

---

## Author Comment (AC2) · 1 Jul 2020

**Response to Reviewer #2**

First of all, I would like to thank this reviewer for his or her thoughtful comments and the potential simplifications he or she has pointed out.

A crucial statement can be found towards the end of the first page of the report: "all the developments around the so-called 'exponential sampling' seem rather useless", and a few lines later this reviewer proposes dyadic expansions *ad hoc*.

Let me answer this misunderstanding with the following figure, which should explain why exponential sampling is the foundation of the whole article:

| Basic Building Block | Bernoulli $B(p)$ | |
|---|---|---|
| $\downarrow$ | Ordinary Sampling | **Exponential Sampling** |
| Derived Distribution | Binomial $B(n, p)$ | Weaver $W(n, p)$ |
| $\downarrow$ | (Standardized) Limit | |
| Limit Distribution | Normal $N(0, 1)$ | Weaver's hem $W(p)$ |

From a combinatorial point of view, there's Pascal's triangle on the left-hand side, and the multiplicative 'triangle' introduced in the manuscript on the right-hand side. From a dynamic system perspective, the paths split and merge on the left, but they only split on the right, which is a crucial feature of chaos and turbulence (with vortices, eddies or boxes multiplying, but not fusing).

Moreover, starting with two constant random variables $X_0 \equiv 0$ and $X_1 \equiv 1$, exponential sampling readily *implies* dyadic expansions. The latter expansions are also crucial for (non-constant) random variables $X_0, X_1$, such that $EX_0 \neq EX_1$, since the expected values can be treated (w.l.o.g) in the same way. Building on this structure, the logical - and by no means 'abrupt' - next step is the general treatment in Section 6, which explicitly includes the random variables' 'inner variability'.

I assume that this reviewer is extremely familiar with cascades, the corresponding models $(\alpha, \beta, p$ and their variants), and many other details of nonlinear processes. Perhaps that is why he seems to have difficulties in grasping the description of the process in the introduction. In algorithmic terms the model studied is:

   0) There are two populations, and $n = 0$.

   1) Select one of the populations at random

   2) Draw an iid sample of size $2^n$ from that population

   3) $n = n + 1$

   4) Proceed to step 1)

Since the first reviewer and all other readers of this manuscript (at least 20, according to Researchgate) have not had any difficulties at this point, I presume that there is no 'missing information' here.

At least in my understanding, the antonym of complicated is not 'trivial', but simple. Starting with an elementary building block, transparent sampling procedures produce basic distributions with straightforward properties ($B(n, p), W(n, p), \ldots$). At least with the wisdom of hindsight, these properties may appear 'trivial'; however, they nevertheless require proof. Moreover, it turns out that, despite - or rather due to - their simple structure,

the Bernoulli and its descendants are pervasive and can be generalized considerably (see the last section for many potentially interesting 'extensions' of exponential sampling).

Finally, I think most scientists would agree that it is worth the effort to reduce complicated matters to reasonable first principles (see, for instance, Feynman's 'prepare a freshman lecture' test', or Lovejoy and Scherzer 2013, "The weather and climate", Chap. 2). In this vein, the referee rightly criticizes that 'tedious algebra' contributes to the length of the manuscript, which the author would be glad to shorten, rectify and *simplify* (see my response to Reviewer #1).

---

## Editor Comment (EC1) · Daniel Schertzer (Editor) · 3 Jul 2020

Dear Uwe,

Both referees provided numerous, constructive comments with detailed explanations, suggestions and clarification requirements. In agreement with their comments, they recommend respectively a rejection and a major revision of your paper. In the latter case, this revision is considered as "quite challenging", because it would require a thorough revision addressing all the comments, starting with the clarification of which "p-model" you consider, and to considerably simplify most the mathematical derivations.

The revised version should be accompanied by careful point by point answers to referee comments. In this respect, the present replies to the referees are far from satisfactory and do not to point to this direction. However, I leave open the possibility of a major

revision, but I do not want to hide that it will require a very careful work, in particular of clarification. Indeed, referees are both bringing into question the relation between the evoked physical topics (multiplicative cascades) and the studied mathematical model.

Best regards, Daniel